# Genome Wide Analysis Points towards Subtype-Specific Diseases in Different Genetic Forms of Amyotrophic Lateral Sclerosis

**DOI:** 10.3390/ijms21186938

**Published:** 2020-09-21

**Authors:** Banaja P. Dash, Marcel Naumann, Jared Sterneckert, Andreas Hermann

**Affiliations:** 1Translational Neurodegeneration Section “Albrecht-Kossel”, Department of Neurology, University Medical Center Rostock, University of Rostock, 18147 Rostock, Germany; Banaja.Dash@med.uni-rostock.de (B.P.D.); Marcel.Naumann2@med.uni-rostock.de (M.N.); 2Center for Regenerative Therapies Dresden (CRTD), Technische Universität Dresden, 01069 Dresden, Germany; Jared.Sterneckert@tu-dresden.de; 3German Center for Neurodegenerative Diseases (DZNE) Rostock/Greifswald, 18147 Rostock, Germany; 4Center for Transdisciplinary Neurosciences Rostock (CTNR), University Medical Center Rostock, University of Rostock, 18147 Rostock, Germany

**Keywords:** amyotrophic lateral sclerosis (ALS), human induced pluripotent stem cells (iPSC), motorneurons (MN), differentially expressed genes (DEGs), Kyoto encyclopedia of Genes and Genomes (KEGG)

## Abstract

Amyotropic lateral sclerosis (ALS) is a lethally progressive and irreversible neurodegenerative disease marked by apparent death of motor neurons present in the spinal cord, brain stem and motor cortex. While more and more gene mutants being established for genetic ALS, the vast majority suffer from sporadic ALS (>90%). It has been challenging, thus, to model sporadic ALS which is one reason why the underlying pathophysiology remains elusive and has stalled the development of therapeutic strategies of this progressive motor neuron disease. To further unravel these pathological signaling pathways, human induced pluripotent stem cell (hiPSCs)-derived motor neurons (MNs) from FUS- and SOD1 ALS patients and healthy controls were systematically compared to independent published datasets. Here through this study we created a gene profile of ALS by analyzing the DEGs, the Kyoto encyclopedia of Genes and Genomes (KEGG) pathways, the interactome and the transcription factor profiles (TF) that would identify altered molecular/functional signatures and their interactions at both transcriptional (mRNAs) and translational levels (hub proteins and TFs). Our findings suggest that FUS and SOD1 may develop from dysregulation in several unique pathways and herpes simplex virus (HSV) infection was among the topmost predominant cellular pathways connected to FUS and not to SOD1. In contrast, SOD1 is mainly characterized by alterations in the metabolic pathways and alterations in the neuroactive-ligand–receptor interactions. This suggests that different genetic ALS forms are singular diseases rather than part of a common spectrum. This is important for patient stratification clearly pointing towards the need for individualized medicine approaches in ALS.

## 1. Introduction

Amyotrophic lateral sclerosis (ALS) is a lethally progressive and irreversible neurodegenerative disease characterized by selective loss of motor neurons in the motor cortex, brain stem and spinal cord. Patients usually progress to muscular paralysis leading to death within 2–5 years after the onset of clinical manifestation [1,2,3]. About 5–10% of ALS cases are familial (fALS), whereas the remaining 90–95% of cases are classified as sporadic (sALS) [4,5,6]. Several studies in ALS patients and model systems have identified numerous genes in ALS pathogenesis, among which four major genes, chromosome 9 open reading frame 72 (*C9ORF72*), superoxide dismutase 1 (*SOD1*), TAR DNA-binding protein 43 (*TARDBP*) and mutations in fused in sarcoma protein (*FUS*), are largely responsible for genetic forms of ALS [7,8]. These causal genes affect diverse processes, including RNA metabolism, protein misfolding, mitochondrial dysfunction, cytoskeletal abnormalities, impaired axonal transport, inflammation and apoptosis [9,10,11,12,13]. *SOD1* mutations appear to account for approximately 20% of fALS cases and 1–4% of sALS cases [14,15,16]. *FUS*, encoding a DNA/RNA binding protein, plays a multifunctional role in RNA metabolism (transcriptional regulation, splicing, mRNA transport) and DNA repair [17,18,19]. Mutations in *FUS* appear to account collectively for about 5% of fALS cases and up to 1% of sALS cases [20,21,22]. Most of these mutations are localized at the C-terminal region. Despite extensive research, the underlying pathogenesis and mechanisms by which these mutations cause motor neuron degeneration and death are still unclear. Hence, till date, there is no cure for ALS, and therefore understanding the molecular signatures of neurodegeneration in ALS can lead to the identification of potential biomarkers that might improve early diagnosis and aid in the identification of therapeutic targets.

As the study of sALS and fALS has led to great advances in development of novel therapies, the identification of significantly dysregulated genes and the mechanistic pathways involved in these mechanisms is of the utmost importance. Very few studies, however, have employed interactome-based approaches to identify shared and unique functional interactions among the gene products that could help with distinguishing molecular subtypes in ALS. Furthermore, even fewer reports exist comparing datasets from totally independent labs and different differentiation protocols for motor neuron differentiation to validate such results. To extend our understanding of finding different molecular mechanisms and pathways related to *FUS* and *SOD1* mutation in ALS disease, we have performed a comprehensive gene expression profiling study using microarray hybridization of the iPSC-derived MN models from control individuals and compared with those from FUS-ALS and SOD1-ALS patients. In addition, we further analyzed previously published independent datasets containing the genome-wide RNA profiling of iPSC-derived MN samples from patients with *FUS* and *SOD1* mutations and controls. This microarray dataset (GSE10638) [23] retrieved from Gene Expression Omnibus (GEO) [24] was screened to identify differentially expressed genes (DEGs) between ALS and control samples before and after drug treatment in order to find potential genes in ALS pathogenesis. Finally, we did a systematic comparison of our results with the ones independently obtained previously. We created a gene profile of ALS by analyzing the DEGs, the Kyoto encyclopedia of Genes and Genomes (KEGG) pathways, the interactome and the transcription factor profiles (TF) to would identify altered molecular/functional signatures and their interactions at both transcriptional (mRNAs) and translational levels (hub proteins and TFs); that profile withstood validation across the different datasets (including different differentiation protocols etc.). By doing so we provide condensed pathophysiological pathways of SOD1 and FUS-ALS to better understand the biological mechanisms underlying motor neuron disease.

## 2. Results

### 2.1. Analysis of Individual Datasets

We analyzed our gene expression datasets of iPSC-derived MNs at DIV 14 of terminal differentiation (=total DIV 30, Appendix A) since major structural degeneration appeared from DIV 16 of motor neuron differentiation (=total DIV 32) onwards [25], and compared them to the NCBI GEO GSE106382 [23] (FUS- and SOD1-ALS and controls) dataset contained data from ALS patients versus healthy controls (DIV35, also prior neurodegeneration became obvious). From GSE106382 dataset, eight independent samples that met our selective criteria were identified containing gene expression profiling data from FUS-ALS iPSC-derived MN (GSM2836938 and GSM2836939), SOD1-ALS iPSC-derived MN (GSM2836942 and GSM2836943) and healthy donor iPSC-derived MN (GSM2836934, GSM2836935, GSM2836936 and GSM2836937; see methods). Descriptions of the datasets analyzed in this study are presented in Table 1. The overall workflow is presented in Figure 1A.

### 2.2. Identification of DEGs between FUS- and SOD1-ALS iPSC MNs in Both Datasets

To identify the unique mRNA signatures in ALS patients, genome-wide profiling was conducted using iPSC-derived FUS and SOD1 motor neurons from ALS patients and healthy controls. Venn analysis of FUS and SOD1 DEGs across all the datasets (present study vs. Fujimori et al., 2018 [23] (GSE106382)) was performed and we revealed no common genes in both diseases (Figure 1B).

Therefore, we performed independent and combined dataset analysis regarding FUS and SOD1 subjects within the two datasets. In the present study, Venn diagram analysis [26] was performed to determine the shared and unique genes between the FUS and SOD1. A total of 108 and 94 significant DEGs (*p*-value < 0.05 and fold change (FC) difference cutoff ≥ 1.5), in FUS and SOD1 compared with controls, were identified. Analysis of DEGs identified four common genes, including G protein-coupled receptor 50 (*GPR50*), synaptosome associated protein 23 (*SNAP23*), NAD(P)H quinone dehydrogenase 1 (*NQO1*) and nuclear RNA (*LOC105376944*), shared between the two subtypes (Figure 2A). A full list of DEGs is provided in Appendix A.

Similarly, the gene expression of GSE106382 datasets obtained from GEO database detected a total of 884 DEGs in FUS and 1680 DEGs in SOD1 samples (*p*-value < 0.05, FC ≥ 1.5) of ALS patients compared to controls, of which 327 overlapping DEGs were identified for both datasets, which were involved in the biological process related to ALS (Figure 2A).

Furthermore, a Venn diagram analysis was also performed to identify the shared DEGs between FUS- and SOD1-ALS datasets (present study vs. GSE106382). Analysis returned 16 and 16 DEGs that were shared between FUS and SOD1 types, respectively (Figure 2A), thereby providing important information regarding a possible molecular footprint of ALS.

### 2.3. Gene Ontology (GO) and KEGG Pathway Enrichment Analysis of DEGs

To gain further insights into the biological significance of identified DEGs and to better understand the pathogenic aspects of FUS and SOD1-mediated ALS forms, we performed a GO and pathway enrichment analysis of target genes by querying EnrichR database [27]. In the present study, the significant GO terms for the DEGs in FUS were enriched in biological processes (BP), molecular function (MFs) and cellular component (CCs). GO enrichment of DEGs indicated the involvement of genes related to neuron differentiation, neuron/brain development and ion binding activity (Appendix A left; Appendix A). Similarly, GO analysis of DEGs in the GSE106383 dataset showed that they were mostly related to axonogenesis, synapse assembly, neuron development and receptor signaling protein serine/threonine kinase activity (Appendix A right). Analysis and comparisons of the enriched GO terms of the SOD1-ALS in both datasets (present study and GSE106383) revealed that a significant number of genes were mainly involved in regulation of membrane potential, synaptic transmission, metabolic processes, regulation of transport and cellular adhesion biological processes. On the basis of molecular function, the most important DEGs in SOD1 prominently included focal adhesion, neurotransmission receptor activity, tyrosine phosphatase activity, voltage gated channel/ion channel activity and calcium mediated signaling functions (Appendix A lower part). The complete list of all GO terms is in Appendix A.

For the analysis of significantly dysregulated pathways (up- and down-regulated) and biological signatures associated with the DEGs in FUS and SOD1, we performed pathway analysis using the bioinformatics tool EnrichR. In our study, KEGG pathway analysis revealed significant pathways associated (*p* < 0.05) with herpes simplex virus infection (HSV), axon guidance and ABC transporters and circadian entrainment as the major altered pathways in FUS-ALS (Figure 2B). In particular, HSV infection was the most significantly enriched pathway in FUS-ALS patients associated with zinc-finger proteins as the major key involved genes. Notably, the majority of the enriched pathways in ALS patients have already been associated with ALS pathogenesis, such as immune/inflammatory response, T cell activation, viral infection and apoptosis [28,29,30,31,32,33]. The top significant pathways are given in Figure 2B and the full list of all KEGG pathways is given in Appendix A. Similarly, pathway enrichment analysis of FUS-ALS DEGs (GSE106383) identified a circadian entrainment, neuroactive ligand–receptor interaction and HSV infection as the top three major deregulated pathways (Figure 2B, Appendix A), thereby significantly overlapping with our datasets.

The pathway analysis of SOD1-ALS DEGs (in the present study) indicated the involvement of genes related to long term depression, synapse-related neuronal functions and metabolic-related processes (Figure 2B, Appendix A). Similarly, for SOD1-ALS DEGs (GSE106382), major altered pathways included the morphine addiction pathway, synapse related functions, metabolic processes and epidermal growth factor receptor (EGFR/ErbB) signaling pathway (Figure 2B, Appendix A), suggesting the impacts of metabolic requirements and neuronal mechanisms in this disease.

Comparisons of FUS- and SOD1-ALS DEGs (versus healthy controls) in our datasets identified only few common signaling pathways associated with ubiquinone/quinone biosynthesis and SNARE interactions in vesicular transport, whereas the common DEGs (GSE106382) were found to be significantly enriched in amino acid metabolism, p53 signaling pathway and NF-kappa B signaling pathways, indicating that common mechanisms may be involved between these diseases (Appendix A).

In addition, comparison analysis between the respective FUS-ALS and SOD1-ALS DEGs across different datasets (present study vs. GSE106382) was also performed because we wanted to focus on common regulatory mechanisms and/or molecular footprints at the pathway level and to better understand the link and the differences encompassed in the development of the two manifestations of these diseases. Interestingly, KEGG pathway analysis of the common DEGs in FUS-ALS revealed significant pathways enriched in HSV infection, metabolism and adherens junction, and most strikingly, “herpes simplex infection” was highlighted as the topmost highly enriched pathway in the network, supporting a potential functional link between HSV infection and FUS-ALS pathogenesis (Figure 2C left). Most importantly, the pathway was predominantly enriched in “zinc-finger proteins” (ZNFs), which mediate direct interactions with DNA, RNA, PAR (poly-ADP-ribose) and other proteins given its crucial role of mediating a wide range of molecular functions. In past years, ZNFs have been implicated to have an important role in the pathogenesis of several neurodegenerative diseases such as spinal muscular atrophy (SMA), Alzheimer’s and Parkinson’s disease [34,35]. In contrast, the gene expression pattern differed in SOD1-ALS, where most DEGs were associated with the thyroid cancer, glycosphingolipid/proteoglycan biosynthesis, long term depression and carbon metabolism pathways, suggesting the impact of neurometabolic defects on SOD1-ALS pathology (Figure 2C right).

### 2.4. DEGs-Transcription Factor Interaction Analysis

To identify key regulators responsible for the observed patterns in FUS and SOD1 gene expression profiling studies, a transcription factor (TF)-target gene interaction analysis was performed using NetworkAnalyst tool [36]. We focused to infer the most important TFs associated with the complete set of DEGs for each disease subtype. The interaction network was constructed using the ENCODE, JASPER and ChEA databases [37,38,39] showing the regulatory relationships involved at transcriptional level (Appendix A). Shared TFs were illustrated by Venn analysis. In the FUS analysis (present study), we did not identify shared TFs between three databases; however, 29 TFs were identified shared between JASPER and ChEA databases, whereas 13 TFs were identified in the SOD1 by all the databases and FUS shared 11 TFs (SREBF1, CREB1, GATA2, YY1, JUN, CEBPB, STAT1, GATA3, PPARG, RELA and ELK1) with SOD1 in the analysis (Appendix A). Similarly, in the FUS analysis (GSE106382), 30 shared TFs were detected between all the databases, whereas 244 TFs were identified in SOD1 (GSE106382) shared by all the databases and FUS shared 26 TFs with SOD (Appendix A). We next identified 19 shared TFs among the FUS datasets (present study vs. GSE106382), whereas SOD1 datasets (present study vs. GSE106382) shared 13 TFs which may play roles in the regulation of DEGs (Figure 3A). The analysis of TFs-target gene interaction network based on shared DEGs identified by FUS- (Figure 3B) and SOD1-ALS (Figure 3C) (present study, GSE106382) is given in Figure 3B and Figure 3C. The complete list of all TFs was given in Appendix A.

### 2.5. Protein–Protein Interactome (PPI) Analysis

To get further insights in differentially affected signaling pathways, we went on to perform protein–protein interactome analysis (PPI) in STRING database [40] to identify unique hub proteins encoded by the DEGs between FUS and SOD1 and their functional interaction/modules with other associated pathways that may serve important roles in ALS pathology.

FUS-ALS DEGs (108, in the present study) were annotated, matched with the database and used to construct the interactome network. Furthermore, an expanded PPI network/interactome was generated by integrating top 200 known or predicted binding partners to form a tightly connected interactome with 279 nodes representing the proteins and 2517 edges representing the interaction between these proteins (PPI enrichment *p*-value = < 1.0 × 10^−16^) (Appendix A). Cluster (MCL clustering) and topological analysis (degree connectivity and betweenness centrality) clearly revealed that the interactome contained major clusters having maximum interactions with hub nodes/genes, such as *MET*, *PRKG1*, *SNAP23*, *BMP7*, *NR4A2* and *RNASEL*. Among these, receptor tyrosine kinase (*MET*; betweenness centrality = 5899.333; degree = 47) and synaptosome associated protein 23 (*SNAP23*; betweenness centrality = 2725.333; degree = 24) were highlighted as the topmost highly connected nodes/hub genes among the up- and down-regulated DEGs, respectively. Functional enrichment analysis of DEGs identified in the FUS datasets indicated that genes in these modules were significantly enriched in several pathways, which were related to ErbB signaling (EGFR family of tyrosine kinases), axon guidance/SNARE interactions, TGF-β signaling pathway, cell adhesion and mitogen-activated protein kinase (MAPK) pathways, indicating that these proteins may serve important roles in maintaining the whole interactome.

While the analysis of FUS-ALS DEGs identified in the GSE106382 dataset retrieved many significant clusters in the interactome (PPI enrichment *p*-value ≤ 1.0 × 10^−16^, 969 nodes, 6171 edges) with the original seed of 884 annotated DEGs, ribosomal protein L31 (*RPL31*) was the most highly ranked hub gene revealing the major interconnected cluster among the dysregulated genes (Appendix A). The top hub genes with the highest degrees were *RPL31*, (betweenness centrality = 123,172.3.5; degree = 101) and ribosomal protein S4 Y-Linked 1(*RPS4Y1*; betweenness centrality = 70,343.02; degree = 82) followed by phospholipase C beta 1 (*PLCB1*; betweenness centrality = 177,523.9; degree = 82), RAN binding protein (*RANBP2*; betweenness centrality = 173,030.1; degree = 62), proteasome 26S subunit, non-ATPase 10 (*PSMD10*; betweenness centrality = 133,550.6; degree = 61), receptor tyrosine kinase (*MET*; betweenness centrality = 82,690.16; degree = 47) and mitogen-activated protein kinase 9 (*MAPK9*; betweenness centrality = 76,445.33; degree = 42). In addition, functional enrichment analysis indicated that these clusters were associated with the ribosome and viral mRNA translation, RNA transport/spliceosome, circadian entrainment, MAPK signaling, Epstein-Barr virus infection and proteasome pathways in FUS-ALS pathogenesis.

The same interactome analysis was performed for the SOD1 datasets (present study and GSE106382). In the present study, the unique interactome (94 annotated genes, 272 nodes, 3877 edges, PPI enrichment *p*-value = < 1.0 × 10^−16^) of dysregulated genes were significantly enriched in cell cycle, proteasome, SNARE interactions (soluble N-ethylmaleimide-sensitive factor attachment protein receptors), cell adhesion, PI3K-AKT/MAPK, stress response, ER-Golgi network and mitochondrial mitophagy, consistent with a role for SOD1 in ALS pathogenesis (Appendix A). The top hub genes highlighting above functional pathways based on degree centrality measure were cyclin A1 (*CCNA1*; betweenness centrality = 7872; degree = 42), tyrosine-protein kinase receptor ret (*RET*; betweenness centrality = 10,828.5; degree = 38), proteasome 26S subunit, non-ATPase 3 (*PSMD5*; betweenness centrality = 5558; degree = 29), synaptosome associated protein 23 (*SNAP23*; betweenness centrality = 4623; degree = 24) and heat shock protein family A (HSP70) member 2 (*HSPA2*; betweenness centrality = 10,199; degree = 20). Similarly, in the GSE106382 dataset, a total of 1680 annotated genes formed a highly interconnected network (without adding known/predicted interactors) with 1536 nodes connected to 9285 edges. The most highly ranked hub genes among the DEGs were mitogen-activated protein kinase 14 (*MAPK14*; betweenness centrality = 9693.273; degree = 21), E1A binding protein P300 (*EP300*; betweenness centrality = 5882.462; degree = 20), SMAD family member 3 (*SMAD3*; betweenness centrality = 10,455.82; degree = 17), epidermal growth receptor factor (*EGFR*; betweenness centrality = 2990.658; degree = 15) and protein kinase cAMP-activated catalytic subunit alpha (*PRKACA*; betweenness centrality = 3633.342; degree = 15) (Appendix A). Pathway enrichment analysis showed that dysregulated gene clusters were mainly enriched in functional terms related to focal adhesion, MAPK/PI3-AKT, mTOR, Wnt signaling, ER-Golgi transport, cell cycle, ribosome, energy production, amino acid and lipid metabolism and ROS-mediated stress response pathways, suggesting they should work collectively to perform different biological functions. The complete results of all PPI network analyses and the list of binding partners are provided in Appendix A. In both SOD1 dataset analyses, the most enriched protein–protein interaction network identified among the mutual DEGs was associated with metabolic pathways, suggesting a potential preference for metabolism in SOD1-ALS.

Next, we asked ourselves whether the common DEGs across FUS and SOD1-ALS datasets (present study and GSE106382) connect within a protein–protein interaction network to identify common pathways for genetic ALS pathophysiology. The PPI (four annotated genes in the present study; 203 nodes and 2445 edges) represented in the common network for FUS- and SOD1-ALS identified only one hub gene *SNAP23* (betweenness centrality = 276; degree = 24) having a significant number of interacting partners associated with the SNARE interactions in vesicular transport, providing evidence for a conserved functional role of FUS and SOD1 in the vesicle:organelle-mediated membrane trafficking processes during the development of the nervous system (Figure 4A). Taken together, our interactome profiling of FUS- and SOD1-ALS confirmed a small degree of overlap of the respective transcriptional profiles, which is consistent with the diverging functional phenotypes during disease progression. Similarly, in the GSE106382 (FUS and SOD1) dataset, interactome-driven cluster analysis (327 genes annotated, 485 nodes and 5897 edges) showed no significant interactions based on network topology scores, while there were few nodes with good numbers of interacting partners (Figure 4B). From these DEGs, ribosomal protein L31 (*RPL31*; betweenness centrality = 72,337.63; degree = 101), arginine and serine rich coiled-coil 1(*RSRC1*; betweenness centrality = 14,403.61; degree = 79), RAN binding protein 2 (*RANBP2*; betweenness centrality = 79,003.01; degree = 61), receptor tyrosine kinases (*MET*; betweenness centrality = 55,859.55; degree = 47, RET; betweenness centrality = 54,128.26; degree = 38), cytochrome C, somatic (*CYCS*; betweenness centrality = 32,772.78; degree = 27) and DExD/H-box helicase 58 (*DDX58*; betweenness centrality = 20,557.75; degree = 19) were among the pathways with the maximum interactions and were associated with RNA polymerase-based transcription, DNA/RNA metabolism, metabolic pathways, Kaposi’s sarcoma-associated herpesvirus infection and cell adhesion respectively. Of note, SNARE interaction is shared by both datasets (compare Figure 4A,B).

Finally, we hypothesized that common DEGs across different FUS and SOD1-ALS datasets connect within a protein–protein interaction network to identify common pathways valid across totally independent datasets, thereby increasing the likelihood that they play a central role in the respective ALS pathophysiology. Network analysis of 16 common DEGs across the different FUS datasets (present study vs. GSE106382) identified no significant interactions in the network itself; however, an expanded interactome was generated by adding a total of 200 known or predicted interactors into the network (PPI enrichment *p*-value = <1.0 × 10^−16^). A total of 215 nodes (genes/proteins) formed a tightly connected network with 2088 edges (Figure 4C), indicating some connectivity enrichment might be expected with maximum interactions. In the study, receptor tyrosine kinase (*MET*), MAPK family signaling cascades (*MAPK1/MAPK3*), phosphatidylinositol 3-kinase (*PIK3CA*) and tripartite motif containing 28 (*TRIM28*) were represented as the most highly ranked hub genes/central nodes showing maximum numbers of interactions with their binding partners and the functional annotations of the genes/proteins associated with the interactome (Figure 4C). Our results indicated significant enrichment in DNA-binding transcription factor activity (RNA polymerase II-specific), cell adhesion molecules, SUMOylation, regulation of transcription (DNA-templated), NF-kappa B signaling and mitochondrial biogenesis pathways. On the other hand, interactome analysis of common 16 DEGs in SOD1-ALS (present study vs. GSE106382) identified a tyrosine-protein kinase receptor ret (*RET*; betweenness centrality = 894; degree = 38) network enriched in pathways associated with ErbB signaling, neurotrophin and focal adhesion (Figure 4D). *RET* has been highlighted as the hub gene (213 nodes, 2296 edges, *p* < 1.0 × 10^−16^) based on maximum connectivity with the neighboring proteins/genes. Receptor tyrosine kinases such as EGFR/ErbB and RET are involved in numerous neuronal functions and altered pathways associated with the development of neurodegenerative diseases [41,42,43,44,45,46,47].

## 3. Discussion

The pathophysiology of sporadic and genetic ALS still remains far from being understood. In this context, baring a few microarray studies, not many studies have employed interactome- and pathway-based approaches discriminating the various genetic forms of ALS including FUS and SOD1. Hence, the main aim of this study was to identify dysregulated transcripts and to explore the potential similarities among two common genetic ALS forms (FUS and SOD1). In this study, we first compared all the DEGs together among FUS- and SOD1-ALS datasets (present study vs. GSE106382) and our analysis showed that gene expression profiles of subjects with FUS and SOD1 (present study) did not overlap with those affected by FUS and SOD1 (GEO dataset) (Figure 1B); however, significant overlaps of DEGs were identified when systematically comparing them with their respective or mutual counterparts, suggesting that similar functional and/or molecular changes occur in these disease subtypes. Therefore, we focused to identify DEGs specific or shared in these independent datasets.

We have identified a total of 108 and 94 significant DEGs (*p* < 0.05) between the four FUS and three SOD1 patients vs. four healthy controls using iPSC-derived MNs. A total of only four common DEGs were identified between the FUS and SOD1 samples, and may therefore prove to be critical for neurodegeneration in both diseases and heterogeneity among fALS cases. Furthermore, a comprehensive and systematic screening of FUS and SOD1 transcriptomes utilizing the publicly available microarray data GSE106382 [23], to further analyze the common molecular determinants from 2 FUS-ALS iPSC lines, two SOD1-ALS lines and four healthy controls identified a total of 884 DEGs in FUS samples compared with control samples, and a total of 1680 DEGs in SOD1 samples vs. healthy individuals, including 327 common DEGs shared between FUS and SOD1. The large differences of the amount of DEGs between the two studies most likely comes from the fact that the GSE106382 dataset mainly contains only samples either from one patient (two clones from the same FUS-ALS patients) or most likely the same family (two lines with the same SOD1 mutation, (no further details available)). In contrast, we used independent samples from different families carrying different well known disease causing mutations with technical replicates in most of the iPSC lines, including a second clone (to account for clone to clone differences).

A comparison of our results with the data obtained from GSE106382 analysis followed by analysis of two different microarray gene expression datasets across different phenotypes, yielded a common signature of a total of 16 DEGs across the FUS and SOD1 datasets (present study vs. GSE106382). Put together, the findings mainly highlighted three important results: (i) The existence of heterogeneity across different datasets, which might arise from many reasons, such as the use of different differentiation protocols, different gene arrays, different reprogramming strategies, genetic backgrounds, etc. (ii) Using interaction pathway analysis of first view non-overlapping datasets clearly revealed common pathway involvements. By doing so, our strategy clearly showed that genetic ALS forms do differ more than they resemble each other, suggesting it is not a common disease. Finally, however, comparing datasets within one genetic ALS form revealed central pathways within the respective pathophysiology.

Interestingly, the zinc-finger proteins known to have DNA-binding transcription factor activity were found to lead the functional data sets in FUS-ALS. On the contrary, DEGs involved in SOD1-ALS datasets were distributed in cell cycle, metabolism dysfunction, proteolysis, response to stress/stimuli and impaired axonal transport and apoptosis. The in depth analysis of DEGs via GO enrichment analysis and pathway analysis unfolded the presence of unique DEGs across two FUS-ALS gene expression datasets, involved in neuronal developmental and differentiation. In addition, many other intracellular processes also seemed to be involved (Figure 2).

Strikingly, the pathway analysis of DEGs in FUS-ALS unfolded significant pathways associated with HSV infection, synaptic vesicle cycle, cell–cell adhesion, MAPK, and PI3-AKT signaling with the HSV infection identified amongst not only the most highly ranked pathways for FUS-ALS but also being clearly upregulated. Of note, a recent report showed that in vitro viral infections can exacerbate FUS-ALS phenotypes in iPSC-derived spinal neurons [48]. More notably, transgenic mice expressing human endogenous retrovirus developed a deterioration and death of MNs similar to the ALS phenotype [49,50,51]. Studies have reported increased levels of HHV-6 and 8 in ALS patients [52,53,54], demonstrating a potential link between HSV infection and ALS. Interestingly, FUS negatively regulates Kaposi’s sarcoma-associated herpes virus gene expression, revealing that FUS is a viral restriction factor [55]. Furthermore, there is more and more evidence that human endogenous retroviruses (HERVs) play a significant role in ALS [50]. HERV-K was recently shown to be activated in a subpopulation of patients with sporadic amyotrophic lateral sclerosis (ALS) and the expression of HERV-K or its env protein in human neurons caused retraction and beading of neurites. Excitingly, expression of HERV-K is regulated by TAR (trans-activation responsive) DNA binding protein 43, the main aggregating protein in sporadic ALS, which binds to the long terminal repeat region of the virus [50,56,57]. Thus, viral infections might represent environmental risk factors of and possible contributors to ALS pathogenesis. Interestingly, another overlaying pathway of both FUS-ALS datasets was circadian entrainment (Figure 2) which very much fits a recent report showing that FUS is a regulator of circadian gene expression with remarkable effects if ALS-causing mutations are analyzed [58]. Of note, stress granules—in whose biogenesis FUS is critically involved—seem to be under circadian control by oscillating EIF2α [59].

The zinc finger proteins (ZNFs) with the C2H2-type motif predominantly regulating transcriptional programs in neuronal development have now emerged as signatories worthy of investigations in their roles in neurodegeneration. For instance, zinc finger protein Zfp106 interacts with multiple RNA binding proteins, including ALS associated proteins FUS and TDP43 [60]. It appears plausible that such an interaction might be impaired by *FUS* mutations, which could add to the phenotype of defective DNA damage repair reported previously. [19,25,61]. The interplay of the pathogenic signature proteins associated with FUS-ALS and the binding interactome therein still remains far from being lucidly explained in the context of FUS-ALS.

Towards demystifying this puzzle, we made an attempt through our interactome analysis for ALS-associated protein FUS using STRING database [40], engaging a PPI network employing 108 and 884 DEGs from each individual FUS-ALS dataset. In PPI network (present study), top hub genes such as *MET*, *SNAP23*, *PRKG1* and *BMP7* were identified, showing the highest degree in all significant clusters (Appendix A). MET was at the core of the PPI network and exhibited the highest degree of connectivity, suggesting a role of MET as a potential molecular signature for FUS-ALS. MET, a transmembrane receptor kinase-mediated PI3-AKT and MAPK signaling pathway mitigating the regulation of neuronal development, differentiation, motility, migration of neuronal precursors and survival also linked to ALS pathophysiology [62,63,64,65,66] with the definitive status of MET as a potential marker of FUS-ALS needing to be corroborated with validation studies. Many of the players of innate immunity presumably essential in the pathogenesis of MN injury in ALS, such as *MYD88* and *RNASEL*, have been also revealed too. Collectively, all the hub genes may possess critical roles in FUS-ALS, and thus can be exploited as potential signatures or markers for ALS.

Interestingly, interactome analysis based on 884 total DEGs from the GSE106382dataset exposed numerous functional clusters enriched in pathways associated with ribosome, virus transcription/immune system activation, circadian entrainment, the proteasome, the cell cycle, mitochondrial oxidative phosphorylation (OXPHOS) and MAPK and PI3-AKT signaling. In cluster analysis, genes such as *RPL31*, *RPS4Y1*, *PSMD10*, *MAPK9*, *CDC7*, *MET*, *RANBP2*, *JAK2*, *GNAS*, *PLCB1* and *VAMP8* were unveiled as hub genes with the highest node degree in significant clusters, the topmost clusters being ribosome (*RPSL31*, *RPL35A*), immune system (*JAK2*) and MAPK/PI3-AKT (*MAPK9*) signaling incidentally all associated with FUS-ALS. Interestingly, the ribosome was at the core of the interactome network and exhibited the highest degree of connectivity through interacting with ribosomal RNAs. Moreover, a previous study proposed an essential link between RNA stability and neurodegeneration in ALS by illustrating corresponding reductions in mitochondrial components and a compensatory increase in ribosome synthesis using iPSC-derived MNs, showcasing metabolic reliance of the active motor neurons on protein translation and mitochondrial function [67]. Owing to the observed dysregulation of DEGs, including zinc finger proteins and RNA/ribosome binding proteins known to associate the viral interaction with host and the MAPK cascade, speculatively HSV infection involvement might have a quick fire role in activating MAPK-associated signaling pathways and TFs like ZNFs, thereby triggering inflammatory dependent manifestations seen during the FUS–ALS disease course.

Contrary to FUS-ALS, totals of 94 and 1680 DEGs corresponding to each dataset in SOD1-ALS (present study and GSE106382) with respect to the controls were identified using bioinformatics analysis. Functional enrichment analysis unraveled the involvement of DEGs of SOD1 MNs with neurological system development and neuropeptide/neurotransmitter activity, suggesting their participation in regulation of genes in the nervous system (Appendix A). Furthermore, PPI network analysis of DEGs in SOD1-ALS revealed an interactome with significant functional clusters centered on genes were enriched in tyrosine protein kinase family of proteins, vesicular transport, cell adhesion and MAPK/PI3K-AKT signaling pathways, all of which form the basis for neuronal death or repair (Appendix A). Notably, cell adhesion molecules have been proven to be critical for synaptic/neuronal plasticity and axon-bundle formation [68,69,70]. Thus, our analysis brings forward the emergence of hub genes as potential markers in SOD1-ALS along with providing functional linkages between a wide variety of cellular processes and nervous system development, suggesting that *SOD1* mutations impair axonal defects through the MAPK kinase family [71].

The extended analysis of the current study also reflected the sharing of the four DEGs between FUS- and SOD1-ALS, with the leading gene in the PPI analysis being a SNARE protein regulating vesicle trafficking and membrane fusion dynamics, SNAP23. Surprisingly, divergently to our results, the transcriptional profiles from GSE106383 dataset identified 327 DEGs shared between FUS and SOD1-ALS involved with neurodegeneration. Functional analysis of these DEGs demonstrated a series of enriched functional terms associated with transmembrane receptor protein kinase activity, serine/threonine phosphatase activity, ion transport, bone mineralization, cell matrix interaction and nervous system development possibly afflicting neurodegeneration in both diseases. The sharing of important gene clusters between FUS and SOD1-ALS was also vigilant in the PPI network, with the hub genes showing the greatest commonality, justifying shared functions associated with the ribosome, RNA polymerase, circadian entrainment, Kaposi’s sarcoma-associated herpesvirus infection, the cell cycle, mitochondrial metabolic functions, p53 signaling and MAPK signaling pathways, reinforcing their demonstrated functions in ALS. Indeed, aberrant expression and activation of the p38 MAPK pathway in MNs is thought to be important for ALS progression [64,65,71]. RNA polymerase was at the core of the interactome network showing the maximum connectivity with other RNA binding proteins, suggesting these proteins may be involved in transcription and RNA processing and linked to ALS, including FUS and SOD1 [6]. In line with our findings, interactome analysis of GSE106383 dataset revealed that the ribosome and mitochondrial factors were highly enriched among destabilized transcripts in both FUS- and SOD1-ALS iPSCs, indicating a conserved gene expression pattern of RNA destabilization between different ALS types, and their connections dysregulated protein synthesis and/or ribosomal biogenesis pathways.

Comparison of FUS- and SOD1-ALS datasets (present study vs. GSE106383) revealed 16 DEGs common to the datasets of FUS- and SOD1-ALS getting significantly visibly associated with biological processes and molecular functions, viz., cell–cell signaling, receptor protein tyrosine/kinase activity, synaptic transmission, cell–cell adhesion and ion binding. Remarkable pathways, including HSV infection and cell adhesion pathways, were shared between FUS-ALS datasets (present study vs. GSE106383) and unambiguously, HSV infection. The pathways were enriched for the FUS-ALS datasets and were consistent with those reported in previous studies in transgenic mice and human samples related to neuroinflammation/immune response in ALS pathogenesis [72].

Thus, to sum up, anomalies in protein synthesis machinery and linked ribosomal pathways that signify FUS-ALS pathologies, can in the due course of time, affect RNA processing events in virus–host interactions, eventually causing death of MNs. PPI analysis identified hub genes with highest degrees *TRIM28*, *MET*, *JAK2*, *UBC*, *MAPK1/MAPK3*, *PLK3CG*, *GSK3B*, *PLCG1*, *ZAP70* and *NCOA2* shared between two FUS-ALS datasets, suggesting a critical role of these genes/proteins as potential transcriptomic regulators for FUS-ALS (Figure 4C) that may regulate many cellular activities, including cellular differentiation, proliferation, apoptosis, inflammation and innate immunity. Fascinatingly, the multidomain protein encoded by *TRIM28* mediates transcriptional silencing together with the large family of Kruppel-associated box repression domain-containing zinc finger transcriptional regulators (KRAB-ZNF) and has been well documented in a wide range of vital functions, including gene expression, epigenetic mechanisms and DNA repair, which are relevant to different aspects of neurodegeneration [35,60,73,74,75,76]. Recent studies indicated that TRIM28-based repression occurs in the mammalian brain and regulates the steady state levels and toxicity of α-synuclein and tau via SUMOylation and provides beneficial effects through the transcriptional regulatory networks [73,77]. In addition, the transcription factor was shown to regulate RNA polymerase II promoter (Pol II) through proximal pausing and pause release at a large number of mammalian genes [78,79]. Furthermore, a critical role of *TRIM28* in lytic gene expression during the early primary infection of herpes virus 8 has been identified [80]. Similarly, KRAB-zinc finger protein ZNF568) has also been implicated in the transcriptional repression activity, partially through the recruitment of the corepressor *TRIM28* and may enhance proliferation or maintenance of neural stem cells [81]. Among its related pathways were gene expression and HSV-1 infection. Nevertheless, given the involvement of *TRIM28* and *ZNF568* in transcriptional regulation, apoptosis and DNA repair, follow up mechanistic studies investigating their functional roles will be valuable to understand the implications in FUS-ALS. In contrast to FUS-ALS, analysis of SOD1 DEGs identified a minimal overlap of 16 genes shared across SOD1-ALS datasets (present study vs. GSE106383). In particular, pathway and interactome analysis preceded by GO enrichment analysis revealed a network enriched in pathways associated with synaptic/neuronal functions and metabolism, indicating metabolic dysregulation considered to be a common hallmark of SOD1-ALS. Collectively, commonalities have been exposed between the transcriptional changes seen between FUS- and SOD1-ALS MNs, with disparities observed too in FUS-ALS MNs displaying greater level of changes in genes enriched in MAPK signaling pathway part of signaling signatures in ALS. Dysregulation of protein kinases in MAPK signaling pathway has been shown to contribute to the pathogenesis of several neurological diseases, including ALS [63]. Transcriptomic profiles build up due to enrichment of disease specific alteration of transcription factors representing significant targets in neurogenesis and degeneration have far reaching consequences on disease pathogenesis and progression [82,83,84]. Hence, we studied TF-gene interactions followed by a comprehensive analysis in the pathogenesis of ALS with the sets of DEGs (in the current study and GSE106382) involved for each FUS- and SOD1-ALS using NetworkAnalyst tool [36]. Analysis of TFs identified the most significant central hubs consisting of JUN, STAT3, YY1, PPARG, EGR1, CREB1, GATA3, SREBF2 and ELK1 as common regulators, which were mainly enriched in the viral infection, including PPAR, AMPK and JAK-STAT signaling pathways (Figure 3). Amongst the substantial TFs associated with the DEGs were early growth response 1 (EGR1), signal transducer and activator of transcription 3 (STAT3) and peroxisome proliferator activated receptor gamma (PPARG), which are involved in different neurological diseases [41,46,85,86,87]. The gene encoding target of EGR1 (TOE1, nuclear) protein identified in FUS and TDP-43 adult mouse brain-knock out involved in splicing events came up in our analysis too, linking FUS to RNA splicing events [88,89], but precise mechanistic details in FUS-ALS context are still lacking. Even though PPARG and its roles in mitochondrial biogenesis and cell survival in neurons through PPARG coactivator 1α (PGC-1α), are well documented with the neuroprotective effects of PPARG agonists well worked out in ALS [85,90,91], its role in lipid metabolism and this aspect in FUS-ALS needs attention.

Adding on to the aforementioned results, we identified 19 shared TFs (JUN, STAT3, GATA3, GATA2, CEBPB, SREBF2, STAT1, YY1, HNF4A, RELA, E2F4, CREB1, MEF2A, TFAP2A, ESR1, PPARG, RUNX1 and ELK1) with high degrees between two FUS-ALS datasets (present study vsGSE106382) that are linked to prion diseases, transcriptional regulation, viral infection and PPAR signaling pathways. This finding supports the prion-like domain of FUS driving cytoplasmic protein aggregations in ALS [18,92]. These enriched TFs displayed maximum interactions with their target genes (*GPR50*, *ABCC9*, *ZNF667* and *CALB1*) and could prove relevant to several neurodegenerative diseases. In contrast, analysis of TFs in SOD1-ALS datasets (present study vs. GSE106382) identified many significant TFs and their related target genes (*RET*, *SORL1*, *ATP2B2* and *CDH12*) (Figure 3). From these TFs, CREB1 as a common regulator among SOD1-ALS datasets, indicating CREB1 signaling cascade may be involved in neuronal/synaptic related functions. Interestingly, dysregulated CREB1 pathways have been consistently implicated in Alzheimer’s and ALS [93,94,95]. On the other hand, we identified 11 and 26 common transcriptional regulators between FUS- and SOD1-ALS in our datasets and GSE106382 dataset, suggesting their involvement in many neurological diseases including ALS (Figure 3). Overall, our findings from the TFs analysis are in favor of their specific character in fALS and support PPARG, EGR1 and CFEB1 as important transcriptional regulators in FUS and SOD1-ALS disease processes. Further, in-depth functional studies are needed to confirm these results and to better understand the mechanisms of MN-associated TF-target gene regulation in the pathogenesis of ALS.

Thus, our comprehensive analysis pinpoints that there are very promising associations with a neurotropic virus enrichment pathway like HSV infection, which is unique to the FUS–ALS, whereas the SOD-ALS is inclined towards the metabolic regulation highlighting differential gene regulations governing genetic predispositions with segregated consequences on biological processes they impact in relation to. Thus, our results open up novel datasets and unexplained transcriptomic targets impacting biological pathways in an idiosyncratic manner, thereby affecting pathogenesis of ALS in corresponding ways.

While the present study contributes important insights into the shared genes, pathways and TFs for FUS and SOD1 at the functional level, it would be sensible to highlight the strengths and limitations of the current study. First, our sample was relatively small, purely in silico based and was not integrated map of multiple omic data sets to further analyze complex neurological diseases such as ALS. Although we performed individual normalization for two different datasets independently, the heterogeneity of technical variations in individual studies cannot be removed completely. Further, our analysis on iPSC-derived MN system adapted to artificial cell culture conditions, media, atmospheric oxygen pressure and other possible confounding factors including different microarray platforms, sample preparation/RNA isolation methods, and different clinical strategies based on population studies may bias the results. Therefore, our results should be replicated in other independent and larger cohort of ALS patients-derived MNs but also post mortem tissue analysis, including sporadic forms. Overall, taking the advantage of iPSC model system and by integrating the recent published data (such as the one presented here), our study provides a unique set of data comparing the transcriptomic alterations and/or events found in MNs from FUS and SOD1-ALS patients and controls, revealing conserved signatures/risk factors in viral infection, ribosome/protein translation and cell adhesion processes. From this view, we showed that both genetic and transcriptomic data are useful to guide further investigations to address the impact of virus infection-induced stress granule dynamics in the context of ALS pathology. The conglomeration of microarray technology with simultaneous bioinformatic analysis of the transcriptomic profiles surfaced from RNA data of iPSC-derived MNs from ALS patients has put forth newer molecular signatures and targets to be explored for getting newer insights into ALS pathogenesis and disease progression in parallel.

## 4. Materials and Methods

### 4.1. Patient Characteristics

We included cell lines carrying mutations in *FUS* (R521C and R521L, two clones from each patient) and in *SOD1* (D90A (two clones), R115G (one clone)), which were identified by Sanger sequencing. Three different cell lines from healthy volunteers (Table 2) were used as the controls. All experiments were in accordance with the Helsinki convention and approved by the Ethical Committee of the Technische Universität Dresden (EK45022009, EK393122012) and patients and healthy volunteers gave their written consent prior to skin biopsy. In this study we analyzed 1 independent microarray from ALS patients.

### 4.2. Generation and Expansion of Cell Lines

Fibroblast cell lines were established from skin biopsies obtained from familial ALS patients and healthy controls. In case of the control lines, genetic testing was performed on them and they were only included if this was negative for mutations in the four main ALS genes *C9ORF72, SOD1, FUS* and *TDP43*. The reprogramming procedure to obtain iPSC from fibroblasts and characterization of control iPSC lines was described previously [25].

The generation of human neural progenitor cells (NPC) and motor neurons (MN) was performed as previously reported [25] based on the original by Reinhardt et al. [96,97]. Importantly, the NPC derivation took about 14–21 days (DIV) which were then used as a resource for final MN differentiation, which was initiated by treatment with 1 µM pumorphamine (PMA) in N2B27 and supplemented with 1 µM retinoic acid (RA) on day three in vitro (DIV). To increase the purity of MN enriched cell culture another split was performed on DIV 9. In parallel, the medium constitution was changed: Instead of PMA and RA, 10 ng/µL BDNF, 500 µM dbcAMP and 10 ng/µL GDNF were added to N2B27 ensure neuronal maturation. On DIV 14 we performed immunocytochemistry (see Appendix A) and RNA isolation. Detailed functional and static characterization of the applied cell culture model was previously reported by us [25,98,99]. Thus, analyses were performed at DIV 30–45.

### 4.3. RNA Isolation and Microarray Hybridization

Total RNA was extracted using the RNeasy Kit (QIAGEN, Hilden, Germany) according to the manufacturer’s protocol. The precipitated RNA was dissolved in RNase-free water and RNA quality was checked with the Agilent Bioanalyzer (Agilent Technologies, Santa Clara, CA, USA). Total RNA (100 ng) was reverse transcribed, in vitro transcribed and reverse transcribed in a second cycle such that the resulting sscDNA could be fragmented and terminally labeled with Biotin using the Ambicon WT Expression Kit (AM4411973) (Ambion, Life Technologies Corporation, Grand Island, NY, USA) and the GeneChip Whole Transcript (WT) Terminal Labeling Kit (900671) (Affymetrix, Santa Clara, CA, USA). Hybridization, staining and scanning of the Affymetrix Human Gene 2.0 ST microarrays (HuGene-2_0-st-v1) were performed according to the Affymetrix GeneChip WT terminal Labeling and Hybridization User Manual (P/N 702808 Rev. 4) (Affymetrix, Santa Clara, CA, USA). and the data saved in CEL files.

### 4.4. Gene-Expression Validation Data Set

In this study we analyzed 1 independent microarray dataset from ALS patients. Processed gene expression data were utilized with accession number GSE106382 [23] (Affymetrix Human Genome U133A Plus 2.0 Array) from NCBI’s Gene Expression Omnibus (GEO) database [24], which studied the molecular mechanisms of the fALS/sALS, and/or drug-treated ALS models based on their expression profiles. The GSE106382 dataset containing 24 samples from fALS patients with *FUS*, *SOD1* or *TDP43* mutations, and sALS versus healthy controls, was obtained from iPSC-derived MNs. From this large dataset, we analyzed a subset comprising 4 datasets each for FUS- and SOD1-ALS subtype (GSM2836938, GSM2836939, GSM2836942 and GSM2836943) compared to 4 healthy controls (GSM2836934, GSM2836935, GSM2836936, GSM2836937) that met the inclusion requirements respectively. For gene expression profiling assays, motor neuron populations in neuronal cell culture at approximately 40 days in vitro (DIV) were analyzed in both datasets. We processed and analyzed all the datasets and identified DEGs with *p*-values < 0.05 and log2 absolute values for fold control FC ≥ 1.5 (see below method).

### 4.5. Data Analysis and Identification of DEGs

Data analysis was performed on Affymetrix CEL files using Partek Genomics Suite 7.0 software (Partek Inc., Saint Louis, MO, USA) under the “Gene Expression” workflow menu according to the standard pipeline method. Data were normalized and summarized with the robust multi-array average (RMA) method [100], which included RMA background correction, quantile normalization, log2 base transformation and median polish probe set summarization. The configuration consisted of a pre-background adjustment for GC content. Library files were those specified by Affymetrix. Differential expression was determined by one-way ANOVA to obtain statistically significant genes. After running ANOVA, genes of interest (DEGs) were then rank ordered in terms of their *p*-values according to the Benjamini–Hochberg step-up multiple test correction method (i.e., *p*-value < 0.05 for significance). This general approach to the microarray analysis of Affymetrix expression data is outlined in Partek technical literature. Afterwards, the expression ratios were calculated for each gene and a 1.5 fold change was used as the criterion for significance (FDR < 0.05 and fold change ± 1.5) and exported into an Excel file for further analysis. Similarly, we identified DEGs from the GSE106382 dataset according to the standard procedure like above and analyzed them independently. The flow chart for data analysis is given in Figure 1. Next, the results for DEGs obtained in this study were compared to corresponding samples from the GSE106382 dataset. Venn diagrams depicting intersections of DEGs analysis were made to categorize the data into two groups of different expression patterns using web based tools [26,101]. The data discussed in this publication have been deposited in NCBI’s GEO database under accession number GSE158264.

### 4.6. GO and Pathway Enrichment Analysis of DEGs

We performed gene set enrichment analysis via EnrichR (https//amp.pham.mssm.edu/Enrich/) [27] to identify gene ontology (GO) and pathways of the DEGs. A list of all 1.5-fold differentially expressed mRNAs obtained from PGS analysis was uploaded into EnrichR for analysis. This analysis was performed to provide more information about the biological functions and pathways significantly enriched in genes (up- and down-regulated) by focusing on three organizing principles of GO terms (BP: biological process, MF: molecular function, CC: cellular component) and Kyoto Encyclopedia of Genes and Genomes (KEGG; http://www.genome.jp/kegg) pathways [102]. An adjusted *p*-value < 0.05 was considered as the cut-off criterion for all enrichment analysis.

### 4.7. Protein–Protein Interaction Network and Hub Gene Analysis

To better understand the functional interactions of the DEGs and identify the best candidate genes in ALS disease subtypes, a comprehensive protein–protein interaction (PPI) network of their encoding products was constructed by using the Search Tool for the Retrieval of Interacting Genes/Proteins (STRING, Version 11.0, https://string-db.org/) [40] and analyzed through the NetworkAnalyst tool [36]. The STRING database collects and integrates all of functional associations between the genes/proteins by consolidating known and predicted interaction data derived from sources including database, experimental, co-expression, text mining, co-occurrence, neighborhood and gene fusion with the highest confidence score. The statistical enrichment analysis in STRING indicated that the PPI interactomes were significantly enriched (*p*-value < 0.05) and it has been well documented that clustering algorithms were useful for grouping proteins into functional clusters or modules. Therefore, functional clusters were identified using the Markov clustering algorithm (MCL) [103] provided by STRING database. The “inflation” parameter which defines the precision of the clustering between regions of strong and weak interactions was set to 1.5 (the higher the inflation, the more clusters you obtain with weak interactions). Briefly, the complete list of DEGs (both up- and down-regulated) was imported into STRING V.11.0, and the PPI network was constructed independently for each dataset with the highest level of confidence between interactions (highest score of >0.9) and a maximum number of interactions to 200 (100 interactors for each direct and indirect functional interaction). The resulting interactome network was further divided into subnetworks of proteins, each of which represented potential functional clusters or functional modules (using MCL clustering). To identify the “Hub” proteins, network visualization and topological analysis were performed through NetworkAnalyst tool. It is based on two widely used topological parameters, betweenness centrality and node degree. In the network, gene represents node (protein/gene) and edges represent interaction between nodes. The “degree” of a node indicates number of connections it has to other nodes; the highest the degree of nodes (hub proteins/genes) represents an important biological function. Betweenness centrality defines how close a node is to all other nodes in the network. In this study, the PPI network was analyzed based on these parameters (node degree ≥15 was selected for threshold) to identify highly interacting hub proteins of their functional and biological relevance. For a large number of seed proteins the “zero order network” construction was selected to contain only the original seed proteins that directly interact with each other, preventing the “hairball effect” and allowing for better visualization and interpretation of interaction network.

### 4.8. Gene-Transcription Factor Interaction Analysis

To gain further insights into the transcription factors (TF) of DEGs, we uploaded the complete list of DEGs into NetworkAnalyst for network analysis. The TF binding site data were predicted by three established TF target prediction databases [37,38,39]. In the Encyclopedia of DNA elements (ENCODE) database, the TFs were identified by BETA Minus Algorithm. Only peak intensity signal <500 and the predicted regulatory potential score <1 were used. The ChIP Enrichment Analysis (ChEA) TF targets database inferred from published ChIP-X data. Finally, JASPER database provided curated and non-redundant TF-binding profiles. The TF-gene interaction network was analyzed based on node degree and edge betweenness topological parameters. Venn diagrams [101] depicting common TFs identified with each database were also made.

### 4.9. Statistical Analysis

Raw microarray data were statistically analyzed using PGS software (Partek Inc, St. Louis, MO, USA). All of the gene expression samples presented in this study were designed for 3 biological replicates (mean ± SD, n ≥ 3). DEGs with an FDR, using the Benjamini–Hochberg, below 5%, were considered statistically significant (*P*-value < 0.05), and a strict criterion of log2FC ≥ 1.5 or log2FC ≤ −1.5 was used for selection of genes. For the functional enrichment and PPI network analysis, significantly enriched GO terms, pathways and interactomes were identified using a *p*-value < 0.05 as the cut off value for statistical significance.

## 5. Conclusions

In conclusion, the present study provided a comprehensive bioinformatics analysis as a promising approach for the identification of candidate DEGs characterizing different fALS subtypes, thereby revealing key factors and pathway mechanisms common to both diseases and differences that are exclusively highlighted in one of the two entities. We have analyzed transcriptomic changes that occurred in fALS with *FUS* and *SOD1* mutations using iPSC-derived MNs from the present study and GSE106382 gene expression profiling datasets. We integrated the common DEGs into various bioinformatics tools to find the enriched GO terms, KEGG pathways, hub genes and TFs. Our findings presented in this study suggest that FUS and SOD1 may develop from dysregulation in several unique pathways and HSV infection was among the topmost predominant cellular pathways connected to FUS and not to SOD1. In contrast, SOD1 is mainly characterized by alterations in the metabolic pathways and alterations in the neuroactive-ligand–receptor interactions. GO terms related to adhesion and signaling were the main functions enriched by the shared DEGs. Furthermore, this study emphasizes the potential of interactome analysis as a dynamic framework to provide new insights into the most significant hub genes underlying the shared pathological signatures between the two diseases. Given the involvement of TFs in viral infection, neuroinflammation and apoptosis, follow up functional studies will be necessary to verify our findings and explore the mechanistic impacts of these candidate TF-genes’ regulation on the outcome of the FUS- and SOD1-ALS disease and might provide potential clues for the development of new therapeutic strategies.

## Figures and Tables

**Figure 1 ijms-21-06938-f001:**
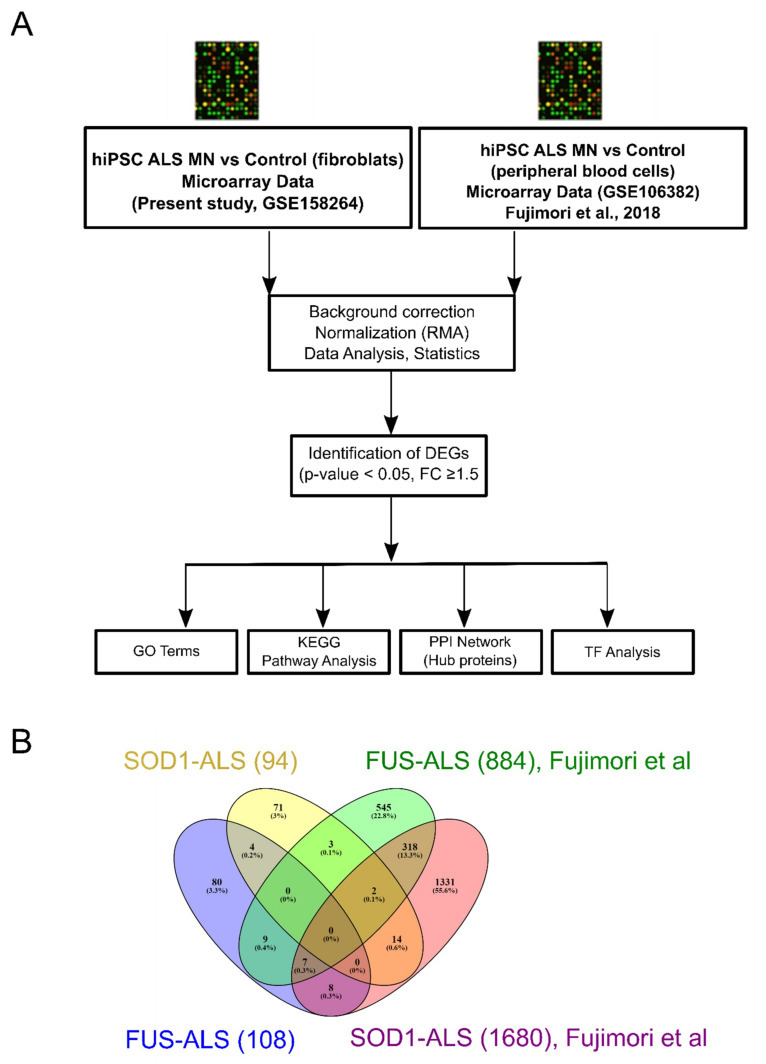
Overall comparison showed no common genes when comparing all datasets. (**A**) Workflow of the study. (**B**) Venn diagram representing all possible combinations of differentially expressed genes (DEGs) across FUS- and SOD1-ALS datasets (present study (GSE158264) and GSE106382). Of note, no common “ALS” DEGs were identified when comparing all datasets.

**Figure 2 ijms-21-06938-f002:**
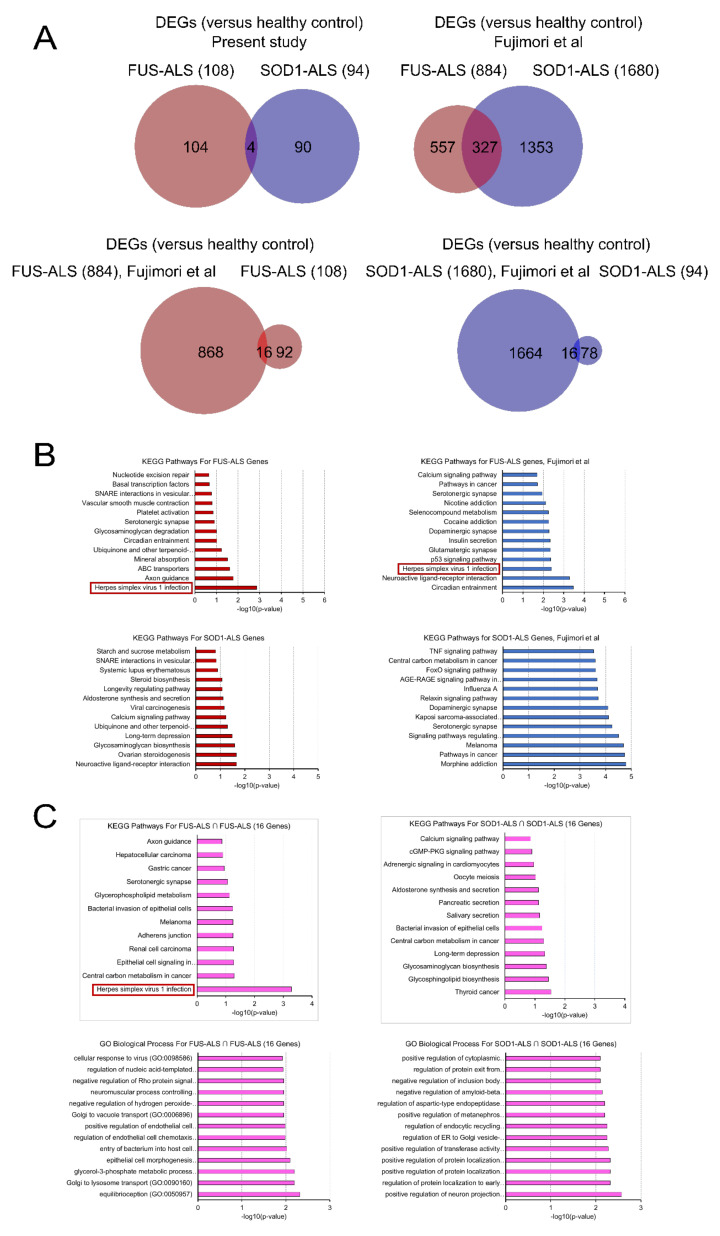
Comparative pathway analysis between the two datasets. (**A**) Venn diagrams showing the number of significantly dysregulated genes in each disease (FUS- and SOD1-ALS) and the observed overlap across comparisons. (**B**) KEGG pathways that are significantly enriched in DEGs specific for each FUS-ALS (upper part) and SOD1-ALS (lower part) disease. (**C**) The significant KEGG pathways (upper part) and Gene Ontology (GO) analysis of biological processes (lower part) enriched by the DEGs shared between FUS-ALS and SOD1-ALS (present study vs. GSE106382) respectively. X-axis represents the statistical significance of the enrichment (-log10(*p*-value)). Color coding represents disease subtypes.

**Figure 3 ijms-21-06938-f003:**
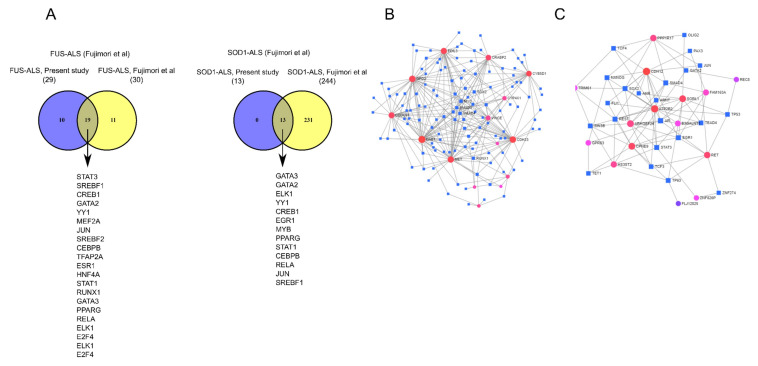
Transcription factor–DEG regulatory interaction network analysis. (**A**) Venn diagrams depicting the TFs specific or common among FUS-ALS and SOD1-ALS datasets. (**B**) Construction of DEG–TF interaction networks based on DEGs identified shared by FUS-ALS datasets (present study vs. GSE106382). (**C**) Construction of DEG–TF interaction networks based on SOD1-ALS datasets (present study vs. GSE106382) and the minimum connected network was analyzed further. The DEG–TF interaction networks were analyzed in NetworkAnalyst tool [36] (i.e., degree of connectivity and betweenness centrality) and statistical significance score *p* < 0.05 was used for the construction of networks.

**Figure 4 ijms-21-06938-f004:**
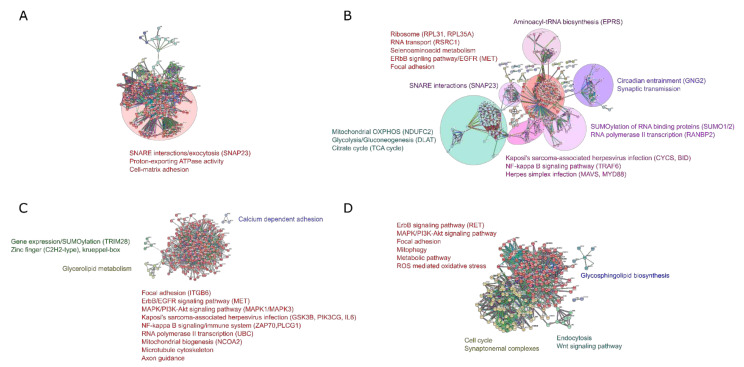
Protein–protein interaction network analysis revealed novel pathways in different genetic ALS forms. (**A**) Protein–protein interaction network of the common DEGs shared by FUS- and SOD1-ALS datasets in the present study. (**B**) Protein–protein interaction network of the common DEGs shared by FUS- and SOD1-ALS in the GSE106382 dataset. (**C**) Protein–protein interaction network of the common DEGs shared by different FUS-ALS datasets (present study vs. GSE106382). (**D**) Protein–protein interaction network of the common DEGs shared by different SOD1-ALS datasets (present study vs. GSE106382). The nodes indicate the DEGs and the edges indicate the interaction between two proteins. The STRING database [40] was used to establish functional associations among the known and predicted proteins using annotated DEGs as queries for FUS- and SOD1-ALS interaction network, with a highest confidence score of >0.9 (STRING scores > 0.900) and a maximum number of interactions of to top 200 (direct and indirect). The interacting proteins have been clustered (MCL clustering, inflation = 1.5) based on their functions and associations to select the most significant functional clusters or sub-networks. To identify the highly interacting hub genes, we visualized the protein–protein interaction network using NetworkAnalyst tool [36] and analyzed the topological parameters of these nodes (node degree ≥ 15). Clusters of functionally related nodes were manually circled and labelled. Disconnected nodes were omitted. The significant hub genes according to degree and betweenness centrality, with the maximum number of connections in the common DEGs network, are highlighted in brackets. The networks having the *p*-values < 0.05 are shown in the network.

**Table 1 ijms-21-06938-t001:** Gene expression datasets selected in this study.

Dataset	Phenotype Type	Brain Region	Platform	Reference
GSE106382	SOD1-ALS (versus Control)	MN	HG-U133_Plus_2	[23]
GSM2836942	SOD1-ALS-patient 1 (H46R-1SOD1-4)	MN	Affymetrix GeneChip Human HG_U133 Plus 2.0	[23]
GSM2836943	SOD1-ALS-patient 2 (H46R-2SOD1-4)	MN	Affymetrix GeneChip Human HG_U133 Plus 2.0	[23]
GSM2836934	Control	MN	Affymetrix GeneChip Human HG_U133 Plus 2.0	[23]
GSM2836935	Control	MN	Affymetrix GeneChip Human HG_U133 Plus 2.0	[23]
GSM2836936	Control	MN	Affymetrix GeneChip Human HG_U133 Plus 2.0	[23]
GSM2836937	Control	MN	Affymetrix GeneChip Human HG_U133 Plus 2.0	[23]
GSM2836938	FUS-ALS-patient 1 (H517D-FALS2e3))	MN	Affymetrix GeneChip Human HG_U133 Plus 2.0	[23]
GSM2836939	FUS-ALS-patient 1 (H517D-FALS2e23)	MN	Affymetrix GeneChip Human HG_U133 Plus 2.0	[23]
In this study; GSE158264	FUS-, SOD1-ALS (versus Control)	Spinal MN	HuGene-2_0-st	
In this study; GSE158264	FUS-ALS-patient 1 (R521L-clone 1)	Spinal MN	Affymetrix GeneChip Human Gene 2.0 ST Array	
In this study; GSE158264	FUS-ALS-patient 1 (R521L-clone 2)	Spinal MN	Affymetrix GeneChip Human Gene 2.0 ST Array	
In this study; GSE158264	FUS-ALS-patient 2 (R521C-clone 1)	Spinal MN	Affymetrix GeneChip Human Gene 2.0 ST Array	
In this study; GSE158264	FUS-ALS-patient 2 (R521C-clone 2)	Spinal MN	Affymetrix GeneChip Human Gene 2.0 ST Array	
In this study; GSE158264	SOD1-ALS-patient 1 (D90A-clone 1)	Spinal MN	Affymetrix GeneChip Human Gene 2.0 ST Array	
In this study; GSE158264	SOD1-ALS-patient 1 (D90A-clone 2)	Spinal MN	Affymetrix GeneChip Human Gene 2.0 ST Array	
In this study; GSE158264	SOD1-ALS-patient 2 (R115G-clone 1)	Spinal MN	Affymetrix GeneChip Human Gene 2.0 ST Array	
In this study; GSE158264	Control 1 (proband 1-clone 1)	Spinal MN	Affymetrix GeneChip Human Gene 2.0 ST Array	
In this study; GSE158264	Control 2 (proband 2-clone 1)	Spinal MN	Affymetrix GeneChip Human Gene 2.0 ST Array	
In this study; GSE158264	Control 3 (proband 3-clone 1)	Spinal Mn	Affymetrix GeneChip Human Gene 2.0 ST Array	
In this study; GSE158264	Control 4 (proband 2-clone 2)	Spinal MN	Affymetrix GeneChip Human Gene 2.0 ST Array	

**Table 2 ijms-21-06938-t002:** Patient/proband characteristics.

Genotype	Cell Culture Model	Sex	Age at Biopsy (Years)	Mutation	Family History	Age of Disease Onset	Clinical Phenotype	Disease Duration (Months)
Controls	hiPSC							
		female	48	-	-	-	-	-
		male	60	-	-	-	-	-
		female	45	-	-	-	-	-
		female	50	-	-	-	-	-
FUS-ALS	hiPSC							
		female	58	p.R521C *	Pos. for ALS	57	spinal	7
		female	65	p.R521L *	Pos. for ALS	61	spinal	60
SOD1-ALS	hiPSC							
		Male	59	p.R115G	Pos. for ALS	n.a	spinal	n.a
		female	46	p.D90A *	Pos. for ALS	n.a	Spinal	n.a

n.a: not available; * two clones were used in the current study.

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
