# Peer review of "Genome Wide Analysis Points towards Subtype-Specific Diseases in Different Genetic Forms of Amyotrophic Lateral Sclerosis"

_ijms, 2020, doi:10.3390/ijms21186938_

Round 1
Reviewer 1 Report
In the present manuscript entitled “Genome wide analysis points towards subtype specific diseases in different genetic forms of Amyotrophic Lateral Sclerosis”, Banaja P. Dash et al. investigated the molecular signatures of differentiated motor neurons from iPSCs that are established from ALS patients with either SOD1 or FUS mutations as well as those from control individuals. By their extensive analyses of DEGs, KEGG pathways, interactome, and transcription factor profiles, the authors revealed that FUS was associated with the HIV-infection-related pathways, while SOD1 was linked to the metabolic pathways as well as the neuroactive-ligand-receptor interactions. Based on these results, they concluded that different genetic forms of ALS could be regarded as a rather distinct disease spectrum. The overall research design for the informatic analyses of this study is sound. However, there are some concerns that must be addressed.
Major points
1) In this study, the authors newly established iPSC clones from a number of ALS patients and control individuals. Further, from these iPSC clones, they claim that they have successfully differentiated motor neurons (MNs) according to the previously published procedures, and used them as materials to create their expression profiles by microarray. However, in this manuscript, there are no experimental data to explicitly show as to whether the MNs, which they used, are properly differentiated or not. The authors must show concrete biochemical and/or immunocytochemical evidences for the generated materials; i.e., MNs from iPSC clones, at least as supplemental data.
2) In addition, even if in the properly-differentiated MNs, their expression profiles might be changed over time in culture-period-dependent manners. For example, in the cited paper (ref. 24), Fujimori et al. showed that the long-term culture of differentiated MNs exhibited progressive degeneration phenotypes. It is thus quite important to clearly define at what stages of the MNs are used, including those obtained from the publicly-available datasets.
3) The authors used several iPSC clones or datasets obtained from same individuals. It is quite common that every iPSC clone originated from the same person exhibits variable phenotypes and the expression profiles. To properly obtain the DEG data, the variability among clones from a single individual should be minimum. The authors should explain such variabilities in the materials used in this study.
4) The authors finally claim that FUS-ALS and SOD1-ALS are rather singular diseases than part of a common spectrum. However, despite of such observable differences between these two types of ALS, it also seems to be true that there are a few similarities between the authors’ originals samples and publicly-available datasets, even in the same gene with different mutation. The reviewer thinks that the authors’ conclusions concerning the disease spectrum cannot be still generalized at this stage, and thus are overstated. Although the words “points towards” in the title may be appropriate, the authors should modify some assertive descriptions throughout the manuscript.
Minor points
5) Typos: lines 243, “insides” should be “insights”.
Author Response
Reviewer 1
Review Report (Round 1)
Comments and Suggestions for Authors
In the present manuscript entitled “Genome wide analysis points towards subtype specific diseases in different genetic forms of Amyotrophic Lateral Sclerosis”, Banaja P. Dash et al. investigated the molecular signatures of differentiated motor neurons from iPSCs that are established from ALS patients with either SOD1 or FUS mutations as well as those from control individuals. By their extensive analyses of DEGs, KEGG pathways, interactome, and transcription factor profiles, the authors revealed that FUS was associated with the HSVinfection-related pathways, while SOD1 was linked to the metabolic pathways as well as the neuroactive-ligand-receptor interactions. Based on these results, they concluded that different genetic forms of ALS could be regarded as a rather distinct disease spectrum. The overall research design for the informatic analyses of this study is sound. However, there are some concerns that must be addressed.
Response: We very much appreciate this overall positive review!
Major points
1) In this study, the authors newly established iPSC clones from a number of ALS patients and control individuals. Further, from these iPSC clones, they claim that they have successfully differentiated motor neurons (MNs) according to the previously published procedures, and used them as materials to create their expression profiles by microarray. However, in this manuscript, there are no experimental data to explicitly show as to whether the MNs, which they used, are properly differentiated or not. The authors must show concrete biochemical and/or immunocytochemical evidences for the generated materials; i.e., MNs from iPSC clones, at least as supplemental data.
Response: We now present these data in the supplement figure S1. Together with the fact, that these lines have already been published using the same differentiation protocol including electrophysiological characterization (PMID: 31108504; PMID: 29362359; PMID: 26946488) we believe that we can state that we investigated properly differentiated and matured motor neurons from the respective iPSCs.
2) In addition, even if in the properly-differentiated MNs, their expression profiles might be changed over time in culture-period-dependent manners. For example, in the cited paper (ref. 24), Fujimori et al. showed that the long-term culture of differentiated MNs exhibited progressive degeneration phenotypes. It is thus quite important to clearly define at what stages of the MNs are used, including those obtained from the publicly-available datasets.
Response: We very much appreciate this comment and want to apologize that we haven’t been clear enough so far. Obviously, the reviewer is right that the expression profiles will change over time since neurodegeneration becomes more and more obvious the older the cultures become. We in depth characterize this in our recent paper (please refer to Figure 2 of PMID: 29362359). We did all our analysis at DIV 14 of terminal differentiation (=total DIV 30). This is at a stage were neurons not yet show structural phenotypes but first axon trafficking deficits became obvious. Thus, our dataset is very similar to the Fujimori dataset who used a timepoint when neurodegeneration starts to become obvious (see supplemental data of Fujimori et al.). We clearly mention this now in the revised version of the manuscript.
Of note, we carefully also re-checked all the files we included from the external datasets. By this we came across a mistake that we included datasets of motoneuron progenitor cells from the Ichiyanagi paper. We consequently removed all datasets including progenitor cell lines and did an extensive reanalysis of all affected datasets. By doing so we believe that the analysis have become much more stringent.
3) The authors used several iPSC clones or datasets obtained from same individuals. It is quite common that every iPSC clone originated from the same person exhibits variable phenotypes and the expression profiles. To properly obtain the DEG data, the variability among clones from a single individual should be minimum. The authors should explain such variabilities in the materials used in this study.
Response: We agree with the reviewer that there are several dimensions of variability. First, clone to clone variances even though derived from the same donor, and even more differences when clones from different patients are combined. In fact this might also explain the larger amount of DEGs found in the Fujimori paper since they used (i) different clones from 1 patient in case of FUS-ALS and (ii) did not specify SOD1 patients however present datasets carrying the same SOD1 mutation most likely being from the same family in case of SOD1 ALS. In order to avoid such family background results and to be as closest as possible to clinical patient cohorts we included both, different clones form the same individual but also from different unrelated patients. We state this much more clearer in the revised version of the manuscript (see materials).
4) The authors finally claim that FUS-ALS and SOD1-ALS are rather singular diseases than part of a common spectrum. However, despite of such observable differences between these two types of ALS, it also seems to be true that there are a few similarities between the authors’ originals samples and publicly-available datasets, even in the same gene with different mutation. The reviewer thinks that the authors’ conclusions concerning the disease spectrum cannot be still generalized at this stage, and thus are overstated. Although the words “points towards” in the title may be appropriate, the authors should modify some assertive descriptions throughout the manuscript.
Response: We toned down the conclusion in the manuscript accordingly.
Minor points
5) Typos: lines 243, “insides” should be “insights”.
Response: We corrected this mistake.

Reviewer 2 Report
This study provides a comprehensive bioinformatics analysis to identify differentially expressed genes (DEGs) for FUS-ALS and SOD1-ALS. They analyzed their data and other data from public sources. In addition, they also found that HSV infection was among the topmost predominant cellular pathways connected to FUS and not to SOD1. In general, this is an interesting study that focuses on investigating the different mechanisms between FUS-ALS and SOD1-ALS.
Comments.
- FUS-ALS and SOD1-ALS are caused by the mutations in FUS and SOD1, respectively. Since these two diseases are known to be caused by these two genes mutations, what do you need to compare their DEGs?
- It is interesting to know HSV infection was among the topmost predominant cellular pathways connected to FUS-ALS. Are there other studies that show HSV is related to ALS?
Author Response
Reviewer 2
Review Report (Round 1)
This study provides a comprehensive bioinformatics analysis to identify differentially expressed genes (DEGs) for FUS-ALS and SOD1-ALS. They analyzed their data and other data from public sources. In addition, they also found that HSV infection was among the topmost predominant cellular pathways connected to FUS and not to SOD1. In general, this is an interesting study that focuses on investigating the different mechanisms between FUS-ALS and SOD1-ALS.
Response: We deeply thank the reviewer for his enthusiastic overall statement!
Comments.
1) FUS-ALS and SOD1-ALS are caused by the mutations in FUS and SOD1, respectively. Since these two diseases are known to be caused by these two genes mutations, what do you need to compare their DEGs?
Response: While we do understand the reviewer‘s thoughts we do believe that this comparison is necessary. All ALS patients share motor neuron degeneration, thus the hypothesis suggest itself that different genetic isoforms of ALS share common pathways/denominators which end into the common final path of motor neuron demise. This is specifically true since over the last decades all clinical translation went through the SOD1 mouse model (and almost all of them failed in patients). Thus we need to understand whether we can identify common pathways misregulated between different genetic causes of ALS or not. We discussed this in more detailed in the revised version.
2) It is interesting to know HSV infection was among the topmost predominant cellular pathways connected to FUS-ALS. Are there other studies that show HSV is related to ALS?
Response: Indeed, we were also fascinated by this data. As suggested we expand the discussion about this exciting finding in the revised version of the manuscript.

Round 2
Reviewer 1 Report
In the revised manuscript entitled “Genome wide analysis points towards subtype specific diseases in different genetic forms of Amyotrophic Lateral Sclerosis”, the authors most appropriately improved the context of the manuscript by responding to my original concerns. However, there are still some minor concerns that should be addressed.
Minor points
1) Supplementary Tables 1-5 may not be attached and/or unavailable.
2) Some elaborated figures are difficult to see. The reviewer recommends using more larger images; e.g., Figures 3 and 4.
3) Supplementary: There were some unedited letters; i.e., A, B, C, and D, below the legend of Fig. S3.
4) Typos: lines 91, “And” should be “and”.
Typos: lines 183, “DEGS” should be “DEGs”.
Typos: lines 190, “aminoacid” should be “amino acid”.
Typos: lines 276, “42)..” should be “42).”.
Typos: lines 648, “[25,98,99]Thus” should be “25,98,99]. Thus”.
Author Response
Reviewer 1
In the revised manuscript entitled “Genome wide analysis points towards subtype specific diseases in different genetic forms of Amyotrophic Lateral Sclerosis”, the authors most appropriately improved the context of the manuscript by responding to my original concerns. However, there are still some minor concerns that should be addressed.
Response: We deeply thank the reviewer for his positive evaluation and thorough corrections.
Minor points
1) Supplementary Tables 1-5 may not be attached and/or unavailable.
Response: The supplementary tables were attached online as wished by the publisher in zip format. Please note that the supplementary tables are in excel format to provide more options for the interested readers.
2) Some elaborated figures are difficult to see. The reviewer recommends using more larger images; e.g., Figures 3 and 4.
Response: We appreciate this comment and now provide enlareged versions with better resolution. I hope the editor is fine with this solution.
3) Supplementary: There were some unedited letters; i.e., A, B, C, and D, below the legend of Fig. S3.
Response: We appologize and corrected these.
4) Typos: lines 91, “And” should be “and”.
Typos: lines 183, “DEGS” should be “DEGs”.
Typos: lines 190, “aminoacid” should be “amino acid”.
Typos: lines 276, “42)..” should be “42).”.
Typos: lines 648, “[25,98,99]Thus” should be “25,98,99]. Thus”.
Response: We appologize and corrected these.
We look forward to hearing from you.
Yours sincerely,
Andreas Hermann
